# Long-read metagenomics using PromethION uncovers oral bacteriophages and their interaction with host bacteria

Koji Yahara [1✉], Masato Suzuki [1], Aki Hirabayashi[1], Wataru Suda[2], Masahira Hattori[2], Yutaka Suzuki[3] & Yusuke Okazaki[4]

Bacteriophages (phages), or bacterial viruses, are very diverse and highly abundant worldwide, including as a part of the human microbiomes. Although a few metagenomic studies have focused on oral phages, they relied on short-read sequencing. Here, we conduct a long-read metagenomic study of human saliva using PromethION. Our analyses, which integrate both PromethION and HiSeq data of >30 Gb per sample with low human DNA contamination, identify hundreds of viral contigs; 0–43.8% and 12.5–56.3% of the confidently predicted phages and prophages, respectively, do not cluster with those reported previously. Our analyses demonstrate enhanced scaffolding, and the ability to place a prophage in its host genomic context and enable its taxonomic classification. Our analyses also identify a *Streptococcus* phage/prophage group and nine jumbo phages/prophages. 86% of the phage/prophage group and 67% of the jumbo phages/prophages contain remote homologs of antimicrobial resistance genes. Pan-genome analysis of the phages/prophages reveals remarkable diversity, identifying 0.3% and 86.4% of the genes as core and singletons, respectively. Furthermore, our study suggests that oral phages present in human saliva are under selective pressure to escape CRISPR immunity. Our study demonstrates the power of long-read metagenomics utilizing PromethION in uncovering bacteriophages and their interaction with host bacteria.

---

[1] Antimicrobial Resistance Research Center, National Institute of Infectious Diseases, Tokyo, Japan. [2] Laboratory for Microbiome Science, RIKEN Center for Integrative Medical Sciences, Kanagawa, Japan. [3] Laboratory of Systems Genomics, Department of Computational Biology and Medical Sciences, Graduate School of Frontier Sciences, The University of Tokyo, Bunkyo City, Japan. [4] Bioproduction Research Institute, National Institute of Advanced Industrial Science and Technology, Tsukuba, Japan. ✉email: k-yahara@nih.go.jp

Human microbiomes are of enormous interest to researchers[1,2] and have been model systems for studying polymicrobial communities[3,4]. Interspecies networks within the microbiome can modulate energy metabolism pathways and affect human health. The two most diverse human microbiomes are intestinal and oral microbiota, which harbor hundreds of coexisting species, including bacteria and viruses. Among them, bacteriophages (phages), or bacterial viruses, in the intestinal microbiome have received increasing attention over the last decade[5,6], whereas those present in the oral microbiota have been less studied.

With the rise of next-generation metagenome sequencing technologies, human gut virome studies have increased rapidly[7]. For example, there have been more attempts to characterize "healthy gut phageome" since a study in 2016 conducted deep-sequencing of DNA from virus-like particles and revealed the presence of completely assembled phage genomes in 64 healthy individuals around the world[8]. Attempts have also been made to explore the associations between human gut virome alterations and diseases[7,9,10]. However, only a small number of metagenomic studies focused on oral phage communities[5], including a study in 2014 that reported an alteration of virome composition in subjects with periodontal disease[11]. The largest oral virome study was conducted in 2015, which generated and analyzed more than 100 Gb shotgun sequencing data from 25 samples (20 dental plaque specimens and 5 salivary) to primarily explore the phage–bacteria interaction network[12]. More recently, shotgun metagenome sequencing of 3042 samples from various environments, including the human oral cavity, was conducted in a project aimed at uncovering the Earth's virome[13], which achieved an almost 3-fold increase in the metagenome samples[14]. Re-analysis of the Earth's virome data revealed signatures of genetic conflict invoked by the coevolution of phages and host oral bacteria enriched in the human oral cavity[15], suggesting it as an attractive system to study coevolution using metagenomic data. Such metagenomic signs of coevolution have been missed in previous large viromic studies in humans[3].

These previous studies, however, were based on short-read sequencing data generated using Illumina sequencer. However, the short-read assembly approaches do have limitations, particularly in assembly contiguity[16,17]. Specifically, generation of short fragmented assemblies impedes the analysis of genomic context or detection of viral sequences using programs, such as VirSorter[18,19], which requires long genomic fragments with sufficient evidence to warrant a prediction[20]. Several more limitations of the short-read assembly approaches were specifically addressed in recent studies that aimed to overcome them using long-read sequencing[21–23]. One of these studies[22] validated the approach of viral long-read metagenomics via nanopore sequencing using mock communities, and found it to be as relatively quantitative as short-read methods, providing significant improvements in recovery of viral genomes, albeit the high error rates. More recent shotgun metagenomic analyses using nanopore long-reads demonstrated improved assembly contiguity[16,17], with much less fragmented assemblies than were achieved by PacBio sequencing, possibly due to less variable coverage with nanopore sequencing[17], although the sequencing error rates are lower in PacBio compared to nanopore[24].

In this study, we conducted a long-read shotgun metagenomic study using PromethION, a recently developed high-throughput nanopore sequencer, for studying oral phageome. We also used Illumina HiSeq for sequencing the same samples, enabling error correction of contigs assembled from the long-reads. Our analyses integrated the deep-sequencing data of PromethION and HiSeq, and uncovered hundreds of metagenome-assembled viral genomes, including a prophage with enhanced scaffolding and its host genomic context, a *Streptococcus* phage/prophage group and jumbo phages/prophages (with >200 kb); the characteristics of these genes; the metagenomic signs of coevolution indicated that the oral phages can evade CRISPR immunity.

## Results

### DNA amount and metagenome sequencing with low contamination of human DNA

The concentrations of DNA extracted by the enzymatic method[25] from two 1 mL saliva samples taken from four healthy volunteers are shown in Supplementary Data 1. The range of concentrations varied from 24.5 to 94.1 ng/μl (average was 63.8 ng/μl), corresponding to at least 1.2 μg total DNA per sample, which meets the input requirement (1.0 μg) of the ligation sequencing kit for PromethION. The amount of metagenome sequencing data obtained for the different samples of each individual using HiSeq and PromethION varied from 37.2 to 55.5 Gb per sample for HiSeq, and from 38.8 to 90.1 Gb per sample for PromethION (Supplementary Data 2), both of which were analyzed in the following workflow (Supplementary Fig. 1).

Preprocessing of the HiSeq data using the EDGE pipeline[26] discarded 0.15–0.22% reads and trimmed 0.31–0.41% of bases by the initial quality control. It then removed 0.20–0.93% of the filtered reads that were mapped to the human genome, which was unexpectedly small. An additional experiment demonstrated that this occurred because, in our protocol, we used the OMNIgene ORAL kit, after which we performed the enzymatic DNA extraction: we collected two additional 1 mL samples of saliva from three of the four healthy volunteers using the OMNIgene ORAL kit or, as an alternative, the RNA*later* stabilization solution, followed by the same procedures of enzymatic DNA extraction, library preparation, and metagenome sequencing using HiSeq ("2nd experiment" in Supplementary Data 2). The preprocessing of sequence data revealed that the proportion of human reads was 0.08–0.54% when using the OMNIgene kit compared to 28.12–37.57% when using the RNA*later* kit (Fig. 1). The OMNIgene ORAL kit had the lytic activity for the existing cells including human cells. Because human cells appeared to be more easily lysed than the bacterial cells under preservation in the OMNIgene ORAL kit, the amount of DNA released from the lysed human cells is likely higher than that released from bacteria. In the enzymatic DNA extraction protocol, the salivary sample is first centrifuged to harvest non-lysed microbial and human cells and viral particles as pellets, which are then subjected to DNA extraction. In the case of salivary samples collected using the OMNIgene ORAL kit, the first centrifugation step may have separated human cell-derived DNA/RNA in the supernatant from the pellet, so the pellet contained almost no human DNA (Fig. 1). In contrast, cells in the salivary samples collected using RNA*later* were generally not lysed under preservation because of an absence of lytic activity by RNA*later*. Therefore, the pellet obtained by centrifugation contained mostly intact microbial and human cells and viral particles, which were then subjected to enzymatic lysis to extract whole DNA. Indeed, the total amount of DNA collected using the OMNIgene ORAL kit was always lower (<100 ng/μl) than that obtained using the RNA*later* solution (>300 ng/μl) (Supplementary Data 1).

Preprocessing of the PromethION data discarded 17.9–28.2% of bases (9.1–18.3 Gb) by excluding reads with average quality score <7 (Supplementary Data 2), and then removed 0.2–9.7% of the filtered reads that were mapped to the human genome.

### Enhanced assembly statistics using the long-reads

Using a large amount of long-reads (13 kb length on average, Supplementary Fig. 2), we assembled the long-reads using the recently developed

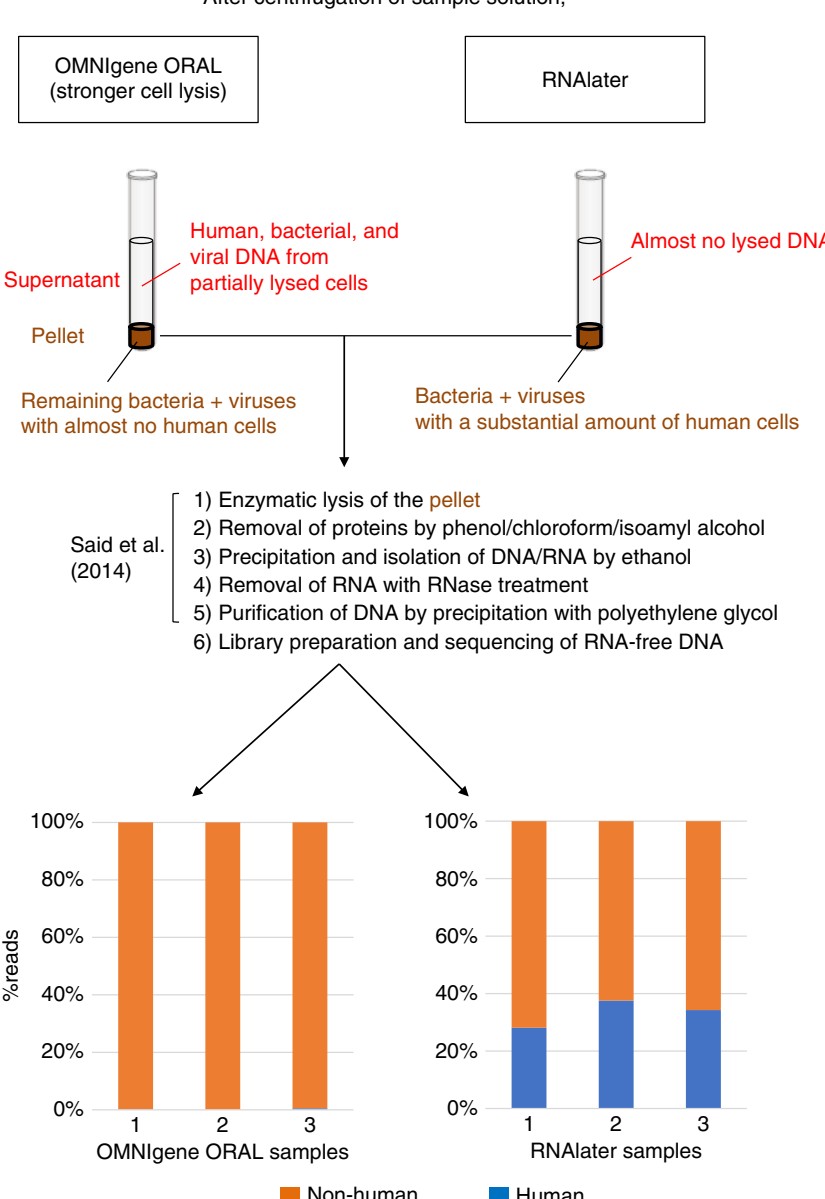

**Fig. 1 Reduction of human reads using the OMNIgene ORAL kit followed by enzymatic DNA extraction.** The scheme at the top outlines the protocol used for DNA collection and extraction and depicts the different constituents of the supernatant and pellet, their separation by centrifugation before enzymatic lysis, and the key steps of the DNA extraction procedure. The bar graph at the bottom shows the percentage of human reads (blue) and non-human reads (orange) present in the three samples stored using OMNIgene ORAL (left) or RNA*later* (right).

assembler Flye[27] with the "–meta" option followed by error correction based on mapping the HiSeq short-reads to assembled contigs. About 94–96% of the HiSeq reads were mapped to the assembled contigs (scaffolds), confirming they well represented the viral diversity in the environment. The number of contigs (≥1 kb, accounting for >99% of all contigs in each sample) after the assembly and error correction ranged from 2865 to 5574 per sample with an average of 3802 (Supplementary Data 3). The N50 ranges from 187 to 345 kb, with an average of 249 kb (Supplementary Data 3). Nucleotide sequences of all the contigs after the assembly and error correction are downloadable at https://figshare.com/s/e211dd1ab1a77ab94e6f. On the contrary, the execution of the hybrid assembly implemented in SPAdes (hereafter hybridSPAdes, for constructing the assembly graph using short-read and followed by gap closure and repeat

resolution using long-read)[28] resulted in a much smaller N50 (11.8 kb on average, in Supplementary Data 3).

**Identified phages/prophages and their taxonomic assignments.** For the contigs after the long-read assembly and error correction, we conducted computational classification using VirSorter and identified phage or prophage genome sequences based on searching for the presence of viral 'hallmark' genes encoding for components found in many virus particles and an enrichment of viral-like genes[18,19]. The phage or prophage regions identified by VirSorter were classified into category 1 ("most confident"), 2 ("likely"), 3 ("possible") phages and category 4 ("most confident"), 5 ("likely"), 6 ("possible") prophages (Supplementary Fig. 3). The "phages" here include various types of viral sequences outside of the main host chromosome, for example,

**Table 1 The number and proportion of viral sequences identified in each sample and stratified by the "most confident" and "likely" phages and prophages.**

| Sample | Phage | | | | Prophage | | | |
|---|---|---|---|---|---|---|---|---|
| | most confident | | likely | | most confident | | likely | |
| | novel | known | novel | known | novel | known | novel | known |
| 1 | 0 (0%) | 5 (100%) | 54 (59%) | 37 (41%) | 26 (46%) | 30 (54%) | 233 (74%) | 83 (26%) |
| 2 | 0 (0%) | 7 (100%) | 37 (49%) | 38 (51%) | 27 (56%) | 21 (44%) | 205 (72%) | 81 (28%) |
| 3 | 7 (44%) | 9 (56%) | 63 (52%) | 58 (48%) | 19 (37%) | 33 (63%) | 323 (77%) | 97 (23%) |
| 4 | 1 (20%) | 4 (80%) | 25 (42%) | 35 (58%) | 3 (12%) | 21 (88%) | 73 (56%) | 58 (44%) |

The novel phages and prophages do not cluster with any viral sequence in the IMG/VR v2.0 database.

extrachromosomal prophages[19]. The number of each viral category identified in this study is shown in Supplementary Fig. 3, and the list of all identified viral sequences is shown in Supplementary Data 4. Due to the difficulty of automatically and reliably excluding false positives from the "possible" candidates, we focused on the "most confident" and "likely" phages and prophages, with the exception of the last section in the "Results" section (the nucleotide sequences are available at https://figshare.com/s/e211dd1ab1a77ab94e6f). We also applied Contig Annotation Tool (CAT)[29] for taxonomic classification of the contigs based on a voting approach by summing all scores from ORFs supporting a certain taxonomic classification (superkingdom, phylum, class, order, family, genus, and species, separately) and checking if the summation exceeds a cutoff value (by default $0.5 \times$ summed scores supporting a superkingdom of certain taxonomic classification across ORFs, which was decided to achieve a balance between the classification precision and fraction of classified sequences). The number of contigs classified as viral by CAT was, however, only 10 across the four samples, all of which were included in those by VirSorter.

In order to examine the novelty of the contigs, for each phage or prophage, we conducted genome clustering[30] with oral viral sequences stored in the largest database (IMG/VR v2.0) of cultured and uncultured DNA viruses[31] developed based on the Earth's virome project using Illumina HiSeq[13]. Those remaining not clustered with a viral sequence in the database (indicated as empty in "clustered with IMG/VR v2.0" column in Supplementary Data 4), were considered as novel, and its number and proportion stratified by the "most confident" and "likely" categories are shown in Table 1. Among the "most confident" phages and prophages, 0–7 (0–44%) and 3–26 (12–46%) were novel, while among the "likely" phages and prophages, 25–54 (42–59%) and 73–323 (56–77%) were novel, respectively.

Furthermore, for each phage or prophage, we conducted taxonomic assignments based on the extent (network) of gene sharing between a query using vConTACT v2.0[32] and the reference viral genome database provided in the software. The genus-level assignment was possible only for 1–6 phages in the four samples (0.2–1.0% of the "most confident" and "likely" phages and prophages), respectively. Family-level assignment (indicated in "assigned family" column in Supplementary Data 4) was possible only for 2–12 (0.9–2.0%) of the known phages clustered with the viral sequences in the IMG/VR v2.0 database, and for 1–3 (0.2–0.9%) of the novel phages (Supplementary Fig. 4), whereas it was possible only for 5–14 (1.3–9.0%) of the known prophages, and for 1–6 (0.3–2.6%) of the novel prophages.

**Enhanced scaffolding and placing a prophage in its host genomic context.** Among the known "most confident" and "likely" prophages clustered with a viral sequence in the IMG/VR v2.0 database, we found cases in which scaffolding and its host

genomic context were much improved compared to the viral sequences assembled from short-reads. An example is shown in Fig. 2a, in which a high coverage prophage with 77.4 kb embedded in a 674.7 kb contig with on average 444× short-read coverage has a genomic region in the middle aligned with 11.1 kb viral sequence in the IMG/VR v2.0 database without information of predicted host. At the end of the high coverage prophage assembled from the long-reads, there is a CDS encoding integrase (depicted as an orange arrow). An enlarged genome map of the prophages is shown in Supplementary Fig. 5, in which genes were characterized by the HMM-based iterative protein searches. Genes for phage morphogenesis (colored in brown) are clustered in the region (approximately from 48 to 60 kb, indicated by a purple horizontal line) corresponding to the aligned region in Fig. 2a. At the right of the end of prophage, there is a CDS encoding enolase (depicted as a green arrow), which is a surface-exposed adhesion protein of *Streptococcus* suggested to be a phage receptor or to interact with proteins of phages[33]. The aligned viral sequence in the IMG/VR v2.0 database encoded 14 CDSs, but its taxonomic classification using CAT was impossible. In contrast, the prophage and host sequences at the both ends were all assigned to *Streptococcus* genus with the score 0.74, 0.90, and 0.79, respectively.

Distribution of differences in length between each of the known "most confidence" and "likely" prophages and corresponding viral sequences in the IMG/VR v2.0 database clustered with the prophages by genome clustering[30] is shown in Fig. 2b. Positive values indicate that the prophages assembled from long-reads are longer than the corresponding viral sequences assembled from the short-reads. The average, median, and interquartile range (IQR) was 47.4, 39.2, and 24.8–60.5 kb, respectively. Only 2.8% (11 out of 393) of the prophages assembled from the long-reads were shorter (i.e., x-axis of Fig. 2b < 0) than the corresponding viral sequence assembled from the short-reads.

***Streptococcus* phage/prophage group and genes for antimicrobial resistance and integrase.** For the high coverage (top 10%) viral sequences in each sample ("top 10% among category 1, 2, 4, 5" in Supplementary Data 4) with expectedly high error correction rate, we conducted a recently developed gene calling (PHANOTATE) specifically designed for phage genomes[34] that are very compact and often have overlapping adjacent genes (10.5–39.4% in the high coverage (top 10%) viral sequences). The iterative protein searches using the HMM-HMM–based lightning-fast iterative sequence search (HHblits) tool that represents both query and UniProt database sequences by profile hidden Markov models (HMM) for the detection of remote homology[35], annotated 53.5% (11,840 out of 22,113) genes. After excluding 33 questionable viral sequences ("top 10% among category 1, 2, 4, 5 (questionable)" in Supplementary Data 4)

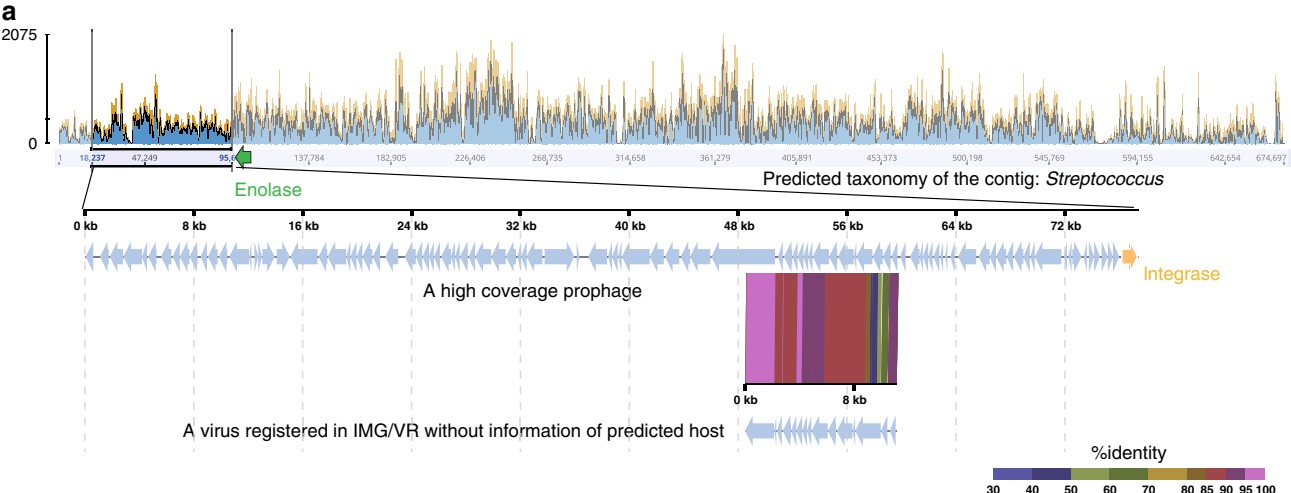

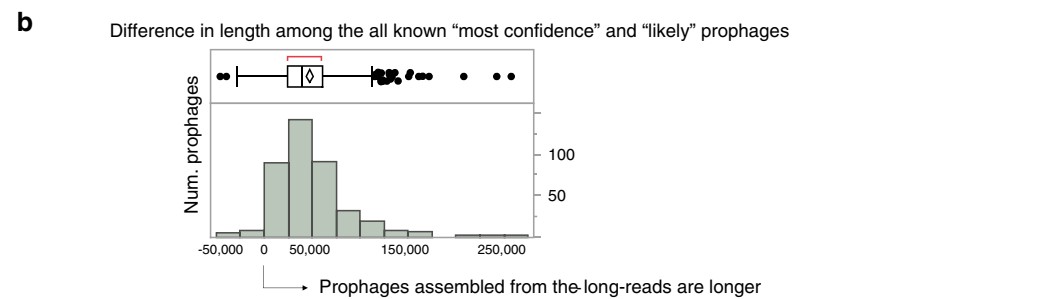

**Fig. 2 Improved scaffolding and host genomic context compared to the viral sequences assembled from short-reads. a** A high coverage oral prophage showing the improvement. Top: coverage plot based on mapping short-reads against it. The prophage region is indicated as a rectangle in the coverage plot spanning the entire contig. Genomic alignment of a high coverage prophage and its viral sequence registered in IMG/VR v2.0 database is shown; the percentage of identity was calculated based on tBLASTx. **b** Distribution of the differences in length between each of the "most confidence" and "likely" prophages and their corresponding viral sequence in the IMG/VR v2.0 database. In the box plot, left and right of the box indicate 25th and 75th percentile, horizontal line indicates median, the middle of the diamond indicates average, the right outliers are above 75th percentile + 1.5× interquartile range, and the red indicates the shortest range in which half of the data distributes. $n = 393$ pairs of prophages and their corresponding viral sequence in the IMG/VR v2.0 database.

without carrying any annotated gene related to phage morphogenesis or transposase, we identified 129 phages/prophages across the four samples, 61.7% of which were novel because they did not cluster with the viral sequences in the IMG/VR v2.0 database. Self-alignment of each viral contig and manual examination of the dot plot confirmed the integrity and absence of redundancy in their assembly. Nucleotide sequences of the 129 high coverage phages/prophages are downloadable at https://figshare.com/s/87a80593aa656ac5567b.

A proteomic tree based on genomic similarity (normalized tBLASTx scores)[36] revealed a notable group (Supplementary Fig. 6) that contained 21.7% (28 out of the 129 across the four samples, Supplementary Data 5) of the high coverage phages/prophages, which was largest among groups that could be seen in the tree. Analysis of the group using a larger proteomic tree, including other reference viral sequences, revealed its location distinctively in a *Siphoviridae* and among *Streptococcus* phages/prophages (Fig. 3). Examination of their genes revealed that 86% (24 out of the 28) of them encoded remote homologs of antimicrobial resistance genes with >99% estimated probability to be (at least partly) homologous to the gene sequences ("CDS related to resistance" Supplementary Data 5). The remote homologs showed an average percentage of amino acid sequence identity of 46% (maximum 99%, minimum 20%, interquartile range 28–61%) and an average percentage of aligned length of 66% (maximum 95%, minimum 13%, interquartile range 47–88%), compared to corresponding amino acid sequences in

UniProt database. Genomic context and coverage of HiSeq read of one of the *Siphoviridae* prophages is shown in Supplementary Fig. 7, in which three remote homologs of antimicrobial resistance genes are colored in red. The remote homologs of genes for phage resistance proteins and acid resistance proteins were also found in 7% and 11% of the phages/prophages in the group, respectively.

The high coverage *Siphoviridae* prophage in Supplementary Fig. 7 is a typical example in which integrase genes are located at the end of the prophage (colored in orange). Similar to the antimicrobial resistance genes, we searched for remote homologs of integrase genes in the group of 28 *Siphoviridae* phages/prophages, and they were detected in 46.4% of them. Average percentage of amino acid sequence identity was 40% (maximum 70%, minimum 21%, interquartile range 25–59%) and average percentage of aligned length was 91% (maximum 100%, minimum 70%, interquartile range 90–95%), compared to corresponding amino acid sequences in UniProt database. If this analysis was extended to all of the high coverage (top 10%) viral sequences predicted by VirSorter, the remote homologs of integrase genes were detected in 52.5% (85 out of 162, indicated in "Integrase" column in Supplementary Data 4) across the four samples.

**Jumbo phages/prophages**. The distribution of the size of the phages and prophages is shown in Supplementary Fig. 8. The

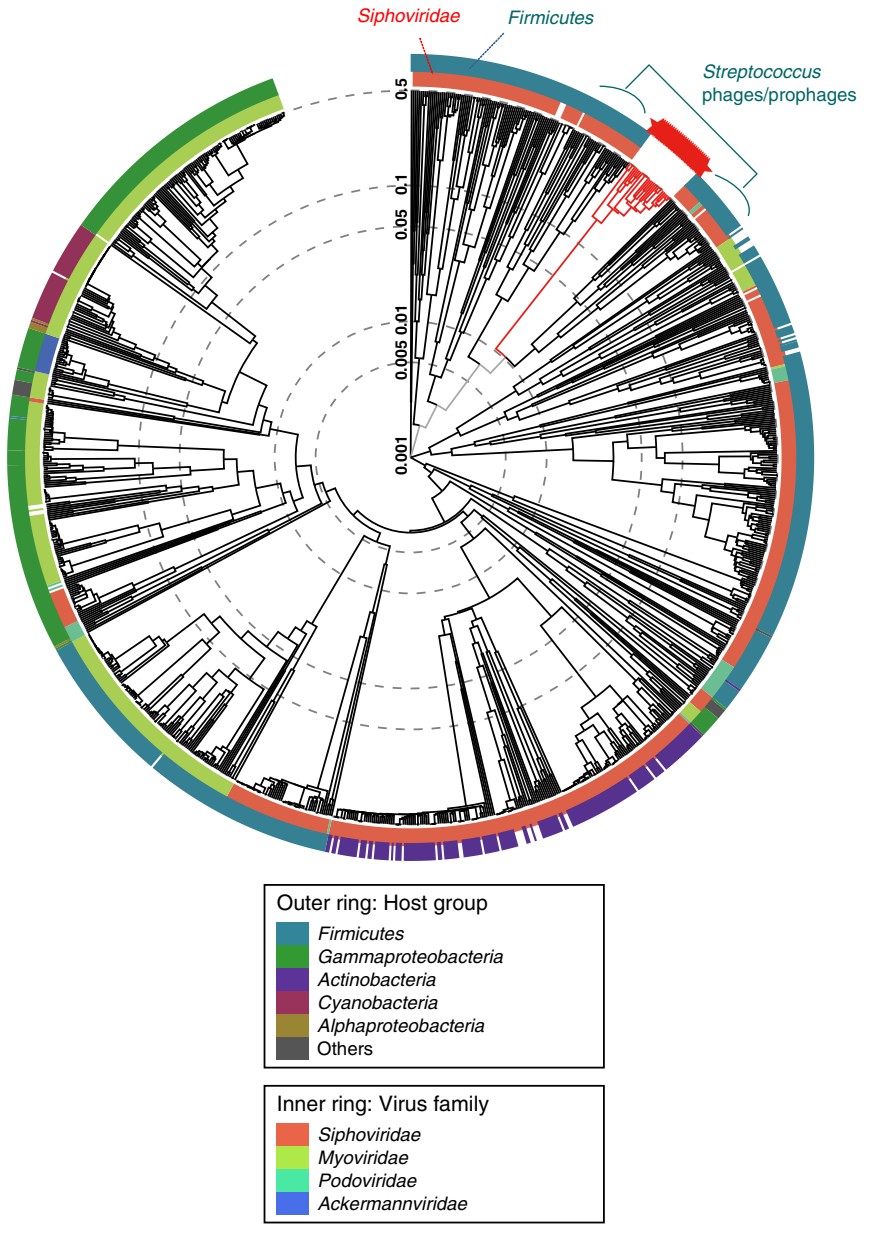

**Fig. 3 Proteomic tree showing a group of high coverage oral phages/prophages.** The red lines at the top right indicates the group consisting of the 28 phages found across the four samples and listed in Supplementary Data 5.

average, median, and interquartile range (IQR) were 57.9, 44.4, 33.9–76.9 kb for the phages and prophages, respectively. We discovered six phages with genomes larger than 200 kb (Supplementary Fig. 8, Table 2), that were classified as jumbo, and had been rarely found but recently began to be identified across Earth's ecosystems[37,38]. We also discovered five jumbo (>200 kb) prophages. Nucleotide sequences of the jumbo phages and prophages are downloadable at https://figshare.com/s/95f9c3cbb074b2782ccf. Manual examination based on self-alignment and dot plots revealed one of them (contig_659 of the third sample) contained a redundant sequence region spanning ~160 kb that might be resulted from an assembly error. After excluding it, 90% of them (nine out of the ten) had at least one phage hallmark gene. One of the remaining contigs with 223,995 bp without a phage hallmark gene (contig_811 in the second sample) was classified as a phage by VirSorter, but 8.9 kb region in the middle (94,241–104,074) encodes three remote homologs of Type IV secretory system protein for conjugative DNA transfer

as well as that of plasmid segregation protein. These results suggested it is rather a plasmid-like element as reported previously[39], although it was not predicted to be a plasmid by a machine-learning program PlasFlow[40], perhaps because of absence of a similar plasmid sequence in the reference training database. Proportion of predicted plasmids among all the predicted viral sequences was 1.9–4.5% ("Plasmid? (PlasFlow)" column in Supplementary Data 4).

For the jumbo phages and prophages, as explained above, we conducted the gene calling specifically designed for phage genomes[34] and found that 13.2–32.6% of the genes were functionally annotated. Regarding the annotated genes, similar to the high coverage oral phages/prophages examined above, remote homologs of antimicrobial resistance genes ("Antimicrobial resistance" in Table 2) were found in 67% (six out of the nine) of the phages/prophages carrying the phage hallmark genes. The remote homologs showed an average percentage of amino acid sequence identity of 35% (maximum 54%, minimum 21%,

**Table 2 Detected jumbo phages and prophages.**

| Sample | Length (bp) | Predicted type | No. of phage hallmark genes | Taxonomy of the contig[a] | Note |
|---|---|---|---|---|---|
| 1 | 263,225 | prophage | 1 | Patescibacteria group: 0.62 Candidatus Gracilibacteria: 0.62 | Encoding a remote homolog of *drrA* that carries out export of the antibiotics |
| 1 | 215,658 | prophage | 6 | Treponema (genus): 0.93 | Fig. S9 |
| 2 | 294,299 | phage | 4 | Selenomonas (genus): 0.59 | Encoding a remote homolog of a beta-lactamase superfamily domain |
| 2 | 270,161 | prophage | 10 | Lachnospiraceae (family): 0.60 | Encoding remote homologs of three metallo-beta-lactamase domain proteins and *vanZ* |
| 2 | 223,995 | phage (plasmid-like element) | 0 | Streptococcus sp. (species): 0.64 | High coverage (top 10%) |
| 2 | 201,060 | phage | 1 | Fusobacterium (genus): 0.86 | Encoding a remote homolog of *drrA* that carries out export of the antibiotics |
| 3 | 217,059 | prophage | 2 | Unclassified Bacteria: 0.50 | Encoding remote homologs of two multidrug-resistant proteins and *drrA* that carries out export of the antibiotics |
| 3 | 201,353 | phage | 5 | Leptotrichia (genus): 0.83 | |
| 4 | 246,016 | phage | 1 | Campylobacter (genus): 0.96 | |
| 4 | 200,581 | prophage | 1 | Leptotrichia (genus): 0.84 | |

aPredicted by Contig Annotation Tool (CAT)[29]; the support values for prediction of the taxonomy of the contig are indicated in parentheses.

interquartile range 29–41%) and average percentage of aligned length of 22% (maximum 67%, minimum 5%, interquartile range 11–28%), compared to corresponding amino acid sequences in UniProt database (see the "Discussion" section).

An example of a jumbo prophage embedded in a bacterial chromosome is shown in Supplementary Fig. 9, in which seven remote homologs of antimicrobial resistance genes with >99% estimated probability are located (reds in Supplementary Fig. 9). Six out of the seven were remote homologs of beta-lactamases and were successively located in a genomic region (~15.5–17.7 kb; Supplementary Fig. 9). The first three were remote homologs of the same beta-lactamase (D0GNG3_9FUSO in UniProt database), which is likely to be an example of the fragmentation of ORF due to the remaining sequence errors on contigs with low coverage (in this case, 29% from the bottom in terms of the rank of coverage). The seventh is a remote homolog of *drrA* (77 amino acid residues were in the pairwise HMM alignment) that carries out the export of the antibiotics[41], which was also found in the other three jumbo phages/prophages (Table 2). The taxonomy of the contig was predicted to be the *Treponema* genus, consisting of dozens of species in the human oral microbiota[42]. Predicted taxonomy of another contig containing a jumbo prophage was *Patescibacteria*, the recently proposed candidate phyla radiation (CPR) lineage that encompasses mostly unculturable bacterial taxa with relatively small genome sizes[43], including ubiquitous members of the human oral microbiota[44]. Overall, there was no overlap in the predicted taxonomies among the jumbo phages/prophages carrying the phage hallmark genes, suggesting they are not confined to specific phylogenetic groups in the human oral microbiota.

**Characteristics of genes encoded in oral phages/prophages.** The pan-genome analysis of all the identified viral sequences to create a gene presence or absence matrix revealed remarkable diversity. Among 115,621 different genes (i.e., the number of rows in the matrix), only 0.3% (309) genes were "core" and present (based on 70% amino acid sequence identity by BLASTp and on average 35% alignment length) in all the four samples, while 86.4% (998,641) genes were singletons (Fig. 4). The amino acid sequence alignments and nucleotide sequences of the 309 core genes are downloadable at https://figshare.com/s/9f76b265f23e23d1e63f. As much as 94.8% (293 out of 309) of the core genes were annotated as hypothetical by a standard annotation program Prokka compatible with the pan-genome analysis[45]. We then conducted the iterative protein searches using the HHblits tool that detected remote homologs with >99% estimated probability for 301 out of the 309 (Supplementary Data 6). A breakdown of their functional categories (Fig. 4) shows 38.9% were homologs of uncharacterized proteins (gray), while the two most dominant annotated functional categories are phage morphogenesis (25.9%, red) and host cellular processes (14.6%, green) such as transcriptional regulators. Of the remaining 20.6%, DNA/RNA metabolism (6.3%, yellow) and phage lysis (3.3%, light purple) accounted for the half, while DNA recombination (light yellow), virulence (purple), nuclease (blue), methyltransferase (orange), prophage anti-repressor (brown), transposase (dark gray), and host's phage susceptibility (khaki) accounted for the remaining half.

**Distribution of CRISPR spacers and its implication to coevolution between oral phages and bacteria.** Finally, for all contigs, we detected CRISPR arrays and spacer sequences. In total, we detected 16,187 unique spacers across the four samples. The number of spacer sequences counted separately in bacterial contigs and phages/prophages (excluding the "possible" sequences) is shown in Fig. 5a. The average number of spacers in

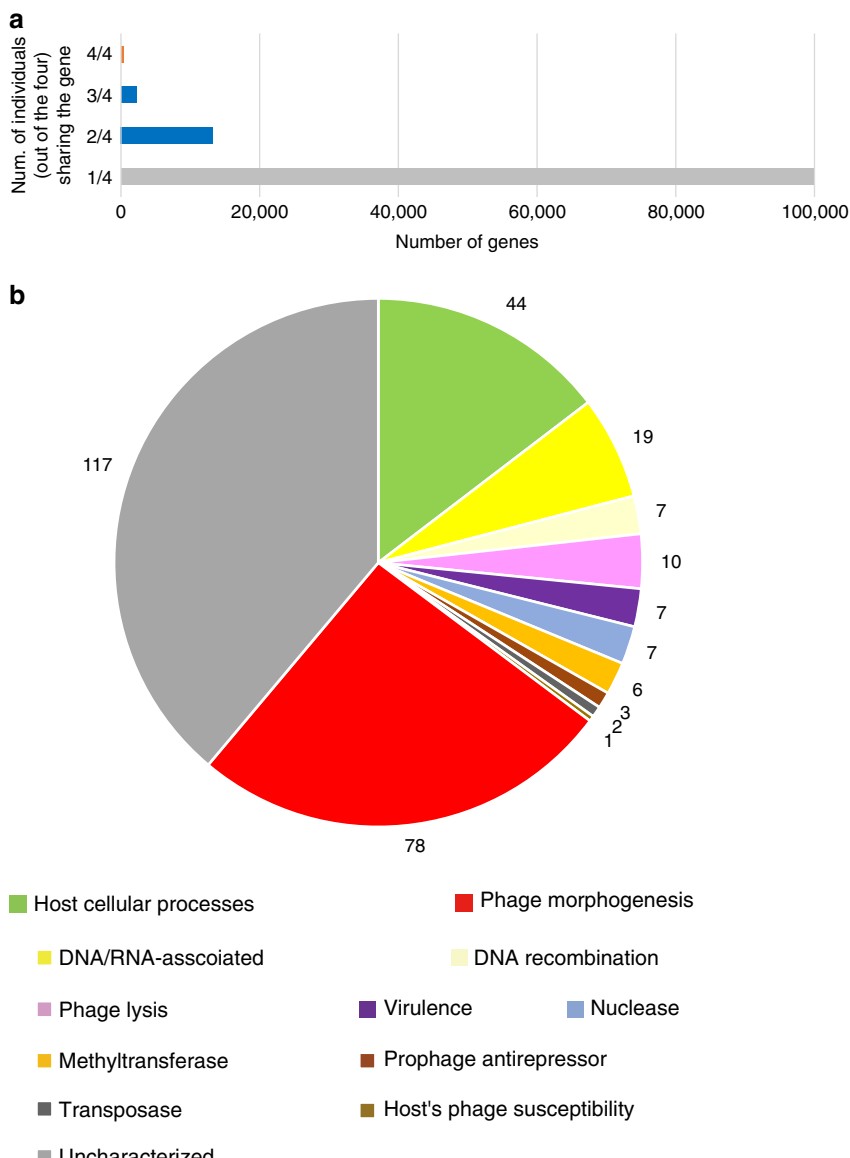

**Fig. 4 Characteristics of genes encoded in oral phages/prophages. a** The number of the gene present in all, three, two, and one out of the four samples. The core (4/4) genes are colored in orange, while singletons are colored in gray. **b** Breakdown of the core genes. The number of genes in each category is shown in the pie chart. More detailed information on each gene is shown in Table S6.

bacterial contigs and prophage sequences was 2503 and 668, respectively (Fig. 5a). Whereas, the spacers in CRISPR arrays were also found in the phage sequences, as reported in other phages[46]: 16 spacers on an average in three samples except for the fourth sample (Fig. 5a). The BLAST search of the spacers in CRISPR arrays located in the phage sequences against two datasets of oral phage/prophage sequences (either in IMG/VR v2.0 database or all the assembled viral contigs in its own sample including the category 3 and 6 ("possible") candidates), revealed almost no homologous sequences ("protospacers") (Fig. 5b, orange). The number was only one or five in the second sample and zero in the other three samples. Whereas, the BLAST search of the spacers in CRISPR arrays in the bacterial contigs and prophage sequences against the IMG/VR v2.0 database revealed that an average 22.2% and 20.3% of the spacers had homologous sequences ("protospacers") in the database, respectively. In contrast, when the BLAST search was conducted against all the assembled viral contigs in each sample, the average proportion decreased to 1.8% for spacers both in the bacterial contigs and in

the prophage sequences. The difference between the two conditions was statistically highly significant ($p < 10^{-15}$, $\chi^2 = 2006.5$, two-sided chi-square test with degree of freedom 1). Further examination of the result of BLAST search of spacers against IMG/VR v2.0 database and that of the genome clustering conducted above (Table 1) revealed that among the oral viral contigs in IMG/VR v2.0 database carrying the protospacers, only 5.2% showed nucleotide similarity enough to be clustered with those identified in the present study, which was significantly lower ($p < 10^{-15}$, $\chi^2 = 579.7$, two-sided chi-square test with degree of freedom 1) than the overall proportion of clustering between the two datasets of viral contigs (36.0%, Table 1). In other words, most of the oral viral contigs in the IMG/VR v2.0 database carrying the protospacers were outside those identified in the present study.

## Discussion

The present study analyzed as much as >30 Gb metagenomic data of both PromethION and HiSeq per sample. The analysis revealed

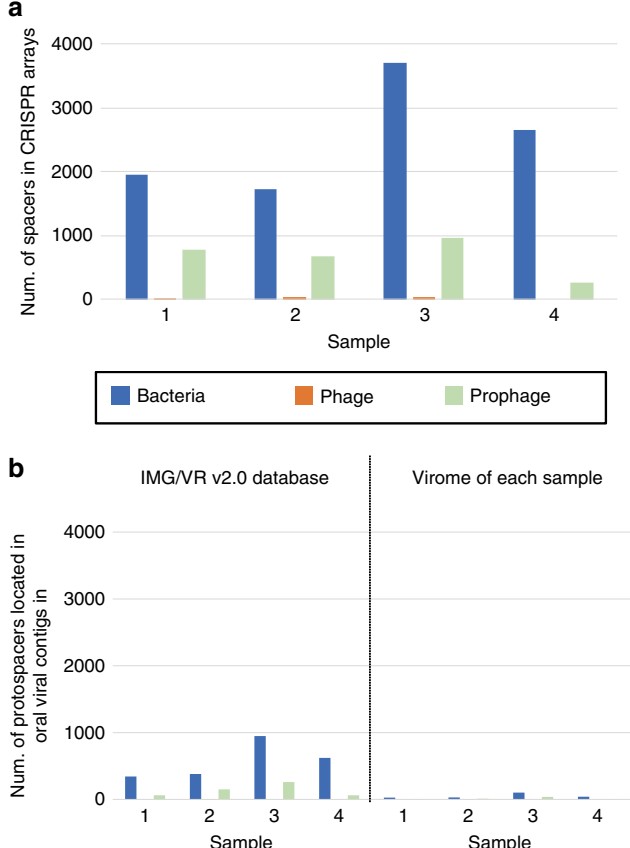

**Fig. 5 Distribution of CRISPR spacers. a** The number of spacers in CRISPR arrays found in the bacterial contig, phage, and prophage regions. **b** The number of protospacers located in oral viral contigs in either the IMG/VR v2.0 database (left) or in those contigs assembled in each sample (right).

40.0–56.3% of the phages and 49.0–72.5% of the prophages were novel (Table 1), and identified as much as 10 jumbo oral phages/prophages (including the plasmid-like element) in an oral environment, significantly increasing our knowledge about "who is there" in the human oral cavity. Such jumbo phages (with >200 kb) were previously found among ~20 host bacterial genus[38,47,48] and recently began to be identified across Earth's ecosystems[37]. However, so far, there has been no publication about these phages in the human oral cavity. Such jumbo prophages were not found among the 12,498 prophages previously identified from publicly available bacterial and archaeal genome sequences using VirSorter[19], in which the longest reported genome sequence did not exceed 140 kb. Furthermore, we demonstrated the long-read sequencing improved scaffolding and ability to place a prophage in the proper host genomic context (Fig. 2) including host genomic sequences at both ends, which enabled taxonomic classification of the contig and deepened our understanding of the interaction between a prophage and its host bacteria. Long-read sequencing using PromethION will boost the discovery of such jumbo phages/prophage more in various environments.

In examining the novelty of the viral contigs assembled from the long-reads, the results of genome clustering[30] with viral sequences in the database should be interpreted with a note of caution for prophages, if their sequences extracted by VirSorter are not completely correct but rather included host genomic sequences. In the high coverage Streptococcus phage/prophage group, 85.7% (24 out of the 28) did not cluster with a viral sequence in the database. In the group, there was a "most confident" phage (contig_2233 in the second sample) that was similar to a known Streptococcus phage

SM1 (NC_004996 in Supplementary Fig. 10) and clustered with seven viral sequences in the IMG/VR 2.0 database (Supplementary Data 4). The genome clustering used cutoffs for obtaining pairwise hits, which are more stringent than the normalized tBLASTx scores used for constructing the proteomic tree. Notably, 21 out of the 24 were "likely" prophages (Supplementary Fig. 6), of which two had genome synteny to the "most confident" phage in the group (Supplementary Fig. 10). Regions outside the synteny of the "likely" prophages did not encode a gene for phage morphogenesis, and might not be viral but host bacterial sequences at both ends mistakenly extracted by VirSorter, making the "likely" prophages not clustered with viral sequences in the IMG/VR 2.0 database. It should be noted that if that is the case, the proportion of novel "likely" prophages, as well as of jumbo prophages, could be overestimated.

The majority (82.8–87.2%) of the "most confident" and "likely" viral sequences identified in our study were prophages integrated in host bacterial contigs (i.e., lysogenic). It has been consistently reported that many phages with temperate lifestyles involving the prophage state are members of the human salivary virome[49], in which the most abundant is Siphoviridae[5,11]. The high coverage Streptococcus phage/prophage group revealed in the present study was indeed phylogenetically in Siphoviridae. Among the 28 viral sequences in the group, 24 (86%) were predicted prophages and the remaining 4 (14%) were predicted phages without surrounding host bacterial sequences. When we checked the percentage of reads mapped on the predicted phages (indicated in "% mapped reads on this contig" column in Supplementary Data 5) for comparison with a recent study of the human gut microbiota[23], we found an average of 0.05% (minimum 0.01%, maximum 0.13%), which is comparable to the highly abundant gut virulent phage, crAssphage (0.03% on average in Japan). Further studies are warranted to experimentally examine whether such predicted phages are actually external to host bacterial chromosome, which will require the enrichment of viral particles and virome sequencing. A previous study reported persistence of phage virions in oral cavities, suggesting occurrence of lysis of their host bacteria even for the temperate phage[50].

A notable feature in oral microbiota revealed from the largest deep-sequencing data of both PromethION and HiSeq was high frequency of the remote homologs of antimicrobial resistance genes; 67% of the jumbo phages/prophages and 86% of the Streptococcus phages/prophages group. There has been controversy as to how prevalent the antimicrobial resistance genes are really in phages, given that the genes are only rarely directly encoded in publicly available phage genome sequences[51]. The remote homologs formed the pairwise HMM alignment not entirely but partly with the antimicrobial resistance genes, and are perhaps unfunctional. However, they might play a role as a source of recombination or horizontal gene transfer to generate a new antimicrobial resistance gene sequence. In the human oral cavity, it is known that the "mosaic" penicillin-resistant genes are generated by recombination among oral streptococci[52] and that the massive diversity of organisms and a large amount of extracellular DNA in oral biofilm matrices are expected to create opportunities of recombination or horizontal gene transfer[4].

The remote homologs in the jumbo phages/prophages showed an average percentage of aligned length of 22% and decreased by 44% compared to that in high coverage (top 10%) phages/prophages in which only the plasmid-like element was larger than 200 kb (Table 2). The low percentage of aligned lengths probably resulted from low read coverages, specifically 1–31× short-read coverages with a median of 5× (except for the plasmid-like element) of these jumbo phage contigs. This could have resulted in insufficient consensus-based nucleotide error correction that generated fragmented ORFs introduced by frameshifts and premature stop codons[53]. Our deep (>30 Gb per sample) long-read metagenomic

sequencing successfully reconstructed the large genomic fragments of rare viruses, but their low-coverage contigs can be erroneous and should be carefully interpreted in further analyses.

Regarding the analysis of the distribution of CRISPR spacers and protospacers, a previous study reported that protospacers were detectable as a small fraction (from 1% to 19%, on average ~7%) of the spacers[54]. The proportion in our study obtained by the BLAST search of the spacers against the oral viral sequences in the IMG/VR v2.0 database was slightly higher than the upper limit (19%). A more recent study of the human gut microbiome[55] reported that the proportion was 23.5% when IMG/VR v2.0 database was used, which is comparable to the findings of our study. When the BLAST search was instead conducted against all the assembled viral contigs in each sample, the proportion was significantly decreased. It suggests that the oral phages currently present in human saliva could be under selective pressure of escaping CRISPR immunity, as suggested in a recent review[4] and study[15], given the hundreds of thousands of CRISPR spacer groups that were transcribed in the human oral cavity[56]. Previously, it was reported that oral streptococcal CRISPR spacers and viruses carrying protospacers coexisted in human saliva of a subject, and CRISPRs in some subjects were just as likely to match viral sequences from other subjects as they were to match viruses from the same subject[57]. Compared to the previous study that examined 3473 unique spacers confined to streptococci of four subjects[57], we expanded the number to >4-fold across various bacterial species. Therefore, the results presented in our study are probably more general and reliable than those obtained in the previous study, with regard to the understanding of the overall relationship between CRISPR spacers and protospacers in the oral microbiota.

As written in the "Introduction" section, a framework of viral, long-read metagenomics via nanopore sequencing (MinION) was recently proposed, and one of its main advantages was the capacity to capture more and longer genomic islands[22]. The genomic island was defined in a previous study by coverage of <20% of the median coverage of the entire contig and >500 bp in length[58]. However, such a decrease in coverage can result from a long read-assembled contig containing a region with too many errors to be corrected via short-read mapping. In particular, long-read assembly of low coverage contig is challenging, and detection of a genomic island in such a contig was not practical. This issue is worth addressing in future studies by exploiting the improvement of per-base accuracy and per-cost throughput of the long-read sequencing platforms.

In this study, we obtained salivary DNA by the enzymatic lysis method[25,59], which is useful for isolation of high molecular weight DNA sufficient for long-read metagenomic sequencing[23]. Unexpectedly, we observed a small percentage of human reads during metagenomic sequencing of DNA extracted from salivary samples stored using the OMNIgene ORAL kit compared with that from the same samples stored using RNA*later*. The difference in quantity of human reads between the OMNIgene ORAL and RNA*later* samples can likely be explained by lytic activity possessed by OMNIgene ORAL but not by RNA*later*. The microbial pellet from OMNIgene ORAL samples contained almost no human cells because they were lysed by undisclosed agents in the kit, and human DNA was separated in the supernatant from the pellet. In contrast, the microbial pellet from RNA*later* samples contained mostly intact microbial and human cells because of the absence of lytic activity in RNA*later* under preservation of saliva.

A remarkable feature we found in the pan-genome analysis was the high diversity of genes encoded by oral phages/prophages: as much as 86.4% of the genes were singletons (i.e., specific to each sample). It is consistent with previous studies of 16S rRNA-based metagenomics reporting individuals' oral microbiota are highly specific at the species level[60]. Our study further demonstrated the

specificity at the gene level based on the very deep shotgun metagenome sequencing.

Furthermore, we conducted identification and HMM-based computational characterization of the core genes of oral phages (most of which were initially annotated as hypothetical) to deepen our understanding of what they are doing. The high fraction of the core genes for host cellular processes after phage morphogenesis supports a recently proposed notion that phages modulate the oral microbiome through multiple mechanisms and represent an additional level of balance required for eubiosis[4]. Experimental characterization of the genes in the future will further deepen our understanding of their functions and roles.

In summary, our study demonstrates the power of long-read metagenomics utilizing PromethION in uncovering bacteriophages with enhanced scaffolding, characteristics of their genes, and their interaction with host bacterial immunity. Our study will provide a solid basis for utilizing PromethION to study bacteriophages and host bacteria simultaneously, and further explore the viral dark matter in various environments.

## Methods

**Saliva collection, DNA extraction, library preparation, and meta-genome sequencing**. Two samples of 1 mL saliva were successively collected and stored using a kit specialized for microbial and viral DNA/RNA (OMNIgene ORAL OM-501) from each of four healthy volunteers (two men and two women aged 35–65 years old). From the four individuals, two saliva samples each were taken and subjected to DNA extraction using an enzymatic method[25]. The extracted DNA samples were stored in 50 μL pure water. From the paired samples per person, we used one with a smaller amount of extracted DNA for Nextera XT library construction and genome sequencing with the Illumina HiSeq 2 × 150 bp paired-end run protocol, and another for library construction using the ligation sequencing kit (SQK-LSK109) and PromethION sequencing. An additional experiment was also conducted in which two saliva samples from three out of the four healthy individuals were collected and stored using either the OMNIgene ORAL kit or an alternative (the RNA*later* stabilization solution, AM7022), followed by the same protocols of DNA extraction, and library construction. The libraries were equally mixed and subjected to a run of multiplex genome sequencing with the Illumina HiSeq 2 × 150 bp paired-end run protocol.

**Preprocessing**. We used EDGE pipeline version 1.5[26] for preprocessing (trimming or filtering out reads, and removal of reads mapped to the human genome) of the HiSeq data. For the PromethION data, we used MinIONQC[61] to check diagnostic plots and NanoFilt[62] ("-q 6 –headcrop 75" option) to filter out reads with average quality <7 and trim 75 nucleotides from the beginning of a read. We then used Minimap2[63] to find and remove PromethION reads mapped to the human genome.

**Assembly, error correction, and quality assessment**. We used Flye[27] with the "–meta and –genome-size 200 m (a value much higher than known bacterial genome size)" option to assemble the preprocessed PromethION long-reads, followed by mapping of the preprocessed HiSeq short-reads to assemble contigs using bowtie2[64] with the "—very-sensitive" option. We then conducted an error correction based on the mapping using a single run of Pilon for each assembled contig (1.87 Gb in total)[65], which was computationally intense and can take several weeks (depending on the number of available CPU cores at that time). We also conducted a hybrid assembly implemented in SPAdes with the "—meta–nanopore" option[28]. Quality assessment for the assembled contigs was conducted using MetaQUAST[66].

**Viral sequence identification, taxonomic classification, coverage estimation, and clustering of assembled contigs**. We used VirSorter[18] with the "-db 2" option using the "Viromes" database. We also used CAT[29] with the default database and taxonomy information created in January 2019 and included in the package. Distribution of length of viral sequences identified by VirSorter was examined using JMP Pro version 13 (SAS Institute, Cary, NC, USA). The coverage of each viral sequence was estimated by FastViromeExplorer[67] using randomly selected 123519885 read-pairs (equal to the number of the least-sequenced sample) from each of the four samples. For high coverage viral sequences (top 10% in each sample), self-alignment and manual examination of the dot plot were conducted using the web interface of NCBI BLASTn. A proteomic tree was constructed using ViPTreeGen, and a larger proteomic tree, including the high coverage viral sequence group and other reference viral sequences, was constructed using ViPTree[36] in which their genome alignment was also visually examined. The viral sequences were clustered with oral viral sequences registered in IMG/VR v2.0 database[31] according to a "Viral genome clustering" procedure included in a nontargeted virus sequence discovery pipeline (from step 8 to 11 in ref. [30], using cutoffs of nucleotide sequence similarity ≥90%, covered length ≥75%, and covered

length requiring at least one contig of >1000-bp length. Taxonomic assignment of the contigs was conducted using vConTACT v.2.0[32] with "–db 'ProkaryoticViralRefSeq85-Merged'–pcs-mode MCL–vcs-mode ClusterONE" option.

**Functional annotation and pan-genome analysis of viral sequences.** The prediction of protein-coding genes in the phages/prophages was conducted using PHANOTATE[34] implemented in multiPhATE[68]. For each predicted gene, we conducted iterative protein searches using HHblits[35], which represents both query and database sequences by profile hidden Markov models (i.e., condensed representation of multiple sequence alignments specifying, for each sequence position, the probability of observing each of the 20 amino acids) instead of single sequences for the detection of remote homology. We used the clustered uniprot20_2016_02 database (http://wwwuser.gwdg.de/~compbiol/data/hhsuite/databases/hhsuite_dbs/), which covers essentially all of the sequence universe by clustering the UniProt database[69] from EBI/SIB/PIR and the non-redundant (nr) database from the NCBI. For all hits with >99% probability of being true positives, we visualized their genomic locations using Geneious software (Biomatters Ltd., Auckland, New Zealand) and individually examined each annotation. We did not include uncharacterized genes in the visualization because the number was too many for the software. Furthermore, we conducted pan-genome analyses using the Roary pipeline with the "-i 70 –group_limit 500000" option[70] after the prediction of protein-coding genes for every contig using Prokka software with the "–kingdom Viruses --metagenome" option[45]. For the core genes, we also conducted the iterative protein searches using HHblits, individually examined each annotation, and manually made the functional categorization.

**Analyses of distribution of CRISPR spacers.** CRISPR arrays were predicted on all contigs using a command-line version of the program CRISPRDetect[71]. For each sample, spacers were extracted from the output files and searched using the BLASTn-short function from the BLAST+ package[72] against either oral viral contigs in IMG/VR v2.0 database or those assembled in this study in each sample. The cutoffs were set with at least 95% identity over the whole spacer length and allowing only 1–2 SNPs at the 5′ end of the sequence, according to a procedure in the Earth's virome project[13].

**Ethics.** This study was approved by the ethics committee of National Institute of Infectious Diseases (approval number 931). We have compiled with all the relevant ethical regulations for work with human participants, and that written informed consent was obtained from the individuals for the sampling procedure and for the use of the samples for research.

**Reporting summary.** Further information on research design is available in the Nature Research Reporting Summary linked to this article.

## Data availability
The data of PromethION and HiSeq after the quality control and removal of human reads were deposited at DDBJ (with JGA accession number JGAS00000000186) and is mirrored at NCBI under BioProject accession PRJDB9452. The data of HiSeq of the additional experiment after the quality control and removal of human reads were deposited at DDBJ and is mirrored at NCBI under BioProject accession PRJDB10605 All nucleotide sequence data provided under the Figshare link were deposited at DDBJ (with accession numbers BNJS01000001-BNJS01004500, BNJT01000001-BNJT01003334, BNJU01000001-BNJU01006278, and BNJV01000001-BNJV01003374 for each sample, respectively). IMG/VR v2.0 [https://img.jgi.doe.gov/vr/] and UniProt [https://www.uniprot.org/] databases were used. Source data are provided with this paper.

## Code availability
The custom code used in this study is available at https://github.com/bioprojects/PromethION-oral-phageome-paper[73]

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

## Acknowledgements
The computational calculations were done at the Human Genome Center at the Institute of Medical Science (the University of Tokyo), and at the National Institute of Genetics. This work was supported by Grants-in-Aid for Scientific Research from the Ministry of Education, Culture, Sports, Science and Technology (MEXT) (19H04846 to K.Y., 16H06429 and 16K21723), Grants-in-Aid for the Research Program on Emerging and Re-emerging Infectious Diseases (JP20fk0108133 to M.S.) from the Japan Agency for Medical Research and Development, and JSPS KAKENHI Grant Number 16H06279 (PAGS). We thank Yutaka Suzuki laboratory members for PromethION sequencing. We thank Ryan Wick and Nick Loman for the instruction of the long-read assembly and thank Matthew B. Sullivan, Yosuke Nishimura, Mart Krupovic, Zara Jennings, Tetsuya Hayashi, Takuro Nunoura, Yoshihiro Takagi, Masaki Shintani, and So Nakagawa for discussions.

## Author contributions
K.Y. and M.S. conceived the study. K.Y. designed the study. K.Y. and Y.O. analyzed the data. M.S., A.H., and Y.S. contributed to the preparation of the materials and metagenome sequencing. M.H. and W.S. contributed to the DNA extraction and directed investigation of the reduction of human reads.

## Competing interests
The authors declare no competing interests.
