## [Peer Review File · Nature Communications]

REVIEWER COMMENTS

Reviewer #1 (Remarks to the Author):

Major comments: The utilization of Kraken as the preferred annotation tool for the short reads is worrisome. Internal and published results have shown that Kraken can increase the number of false positives (1). Kraken also seems to not attempt to discriminate or classify viral sequences (2).

Another major concern is the high error rate with PromethION and related sequencing technologies. While the authors attempt to address this issue in lines 345-353, in order to understand how this error rate is affecting virus discovery from their metagenomic datasets, the authors need to include a virus, or a viral mock community in the analysis, and use the same sequencing technology and annotation tool.

Lines 100-105: The low abundance of human reads in saliva samples is surprising. Can the authors very briefly include in this part of the Results section a description of the method and why they think it reduces so significantly the number of human reads?

Figure 1A and 1B do not have to be one of the main figures. Since the figure is only showing DNA concentration, and number of reads, results can easily be added to two tables. Also, it is not clear from looking at figure 1A what the sample designations are unless you look at the text and understand that these are duplicates.

Information from figure 2A and 2B can also be added into a table for easier interpretation.

Lines 187-188: Can the authors expand on what they mean with remote homologs of antimicrobial-resistance genes throughout the text as needed? Is it hits with <95% identity, for example?

Figure 5 and lines 190-193: It is suggested that this is a prophage, but the highlighted annotation does not show markers of lysogeny, particularly integrases or any genes part of lysogeny modules. Can the authors expand on the detection of integrases, etc. in the presumptive prophages?

Methods: Can the authors specify in the methods if they used a DNase treatment before nucleic acid extraction? If not, it is highly possible that host DNA may have been analyzed along with the viral sequences. That may explain plasmid-like sequences in the presumptive prophage sequenced.

Minor:

Line 264: Replace “the on an” with “that an”

Reference titles:

1. Phylogenetic microbiota profiling in fecal samples depends on combination of sequencing depth and choice of NGS analysis method.
2. VirFinder: a novel k-mer based tool for identifying viral sequences from assembled metagenomic data.

Reviewer #2 (Remarks to the Author):

In this manuscript, Yahara and colleagues perform metagenomic sequencing on oral samples (human saliva) using long read sequencing. Applying this fairly new technology, the authors generate >30Gb data (a combination of nanopore and HiSeq data) and then perform assembly. They queried the assembly data to identify viral contigs and by comparing these data to available published data find that a substantial proportion of the contigs are candidate novel phages. Some of these phages encode potential remote homologs of antimicrobial resistance genes.

Overall, the manuscript presents some nice, new data (oral long-read sequencing) from a very limited number of human samples (four subjects, in total). The focus on the oral phageome is interesting and the bioinformatic analyses are generally appropriate. There isn't quite enough methodological detail to understand how the bioinformatic pieces fit together, and in some cases, more methodological detail is needed to evaluate the choices made (e.g. what kraken database was used and how cutoffs were selected). The identification of novel jumbophages is very interesting and was probably the highlight of the manuscript. While interesting, the manuscript could benefit from improvements in writing flow, figure presentation (both how the figures are presented and what is presented in the main manuscript), and much better referencing of the current status of the field, as well as more accurate representations of what is known in the field. I have outlined several major and minor concerns below, which if addressed, would increase my enthusiasm for this piece of work.

Major concerns:

- In general, the introduction, while interesting, makes some critical misinterpretations or misstatements about the current status of the field or prior literature. There are also some notably omissions. I've highlighted some of them here:

o Line 69 – in the PHASTER paper that the authors refer to, the manuscript cited showed that sensitivity of PHASTER increases to >90% in CONTIGS of >20kb – NOT contigs assembled from reads > 20kb. Thus, a core logical point that the authors point at to justify their application of long reads is flawed.

o Lines 75 onward – many have shown that PacBio sequencing can be performed on as little as 100ng of DNA (example: doi: 10.3390/genes10010062). Certainly gut microbiome-based libraries can be made with 1ug of DNA (see <https://www.nature.com/articles/s41587-020-0422-6>)

o Several papers have come out showing the utility of nanopore sequencing for microbiome studies (none of which are cited in this manuscript) – notably:

☐ <https://www.nature.com/articles/s41587-020-0422-6>

☐ <https://www.nature.com/articles/s41587-019-0202-3>

☐ <https://www.ncbi.nlm.nih.gov/pubmed/31359005>

o I don't think that the statement that nanopore library prep requires less DNA than PacBio is accurate based on the available literature. That said, it is a useful platform and there is plenty of justification to use nanopore.

o In lines 98-99 – how were samples multiplexed for the PromethION sequencing? Did they obtain 38.8Gb to 90.1Gb Per sample? It is not clear to me based on the way the manuscript was written, although based on Figure 1b, it appears that this is the case.

o Figure 1 – this can likely be supplemental.

o Line 102-104 – I agree that the % of human reads is very surprisingly small – how does this compare to other studies that used this extraction protocol? Human cells are typically much easier to lyse than bacterial cells...

o Line 108 – need more detail re what the “full database” for Kraken is. The manuscript that is referred to was published in 2014 and there is a minidatabase that is described with it in addition to larger builds. The databases are constantly being updated and many users create their own custom database. Please provide more detail on the exact database used in the methods.

o The method applied for taxonomic classification described in lines 139 onward is interesting – but how did they validate this approach? Is there precedent for this? How was the cutoff value decided?

o Figure 2 – can be supplemental and probably best represented as a table.

- o Figure 3 – What do the numbers of the virsorter categories represent (1, 2, 4, 5)? I don't understand that. Also, this figure might be better if separated into panels for the pro-phage and phages separately. Or perhaps this might be better as a table instead of a figure.
- o The discovery of the oral microbiome jumbo phages is interesting – table 1 is excellent.
- o I don't find the bottom panel of Figure 4 terribly informative.
- o Figure 5 & 6 should be supplementary.
- o Figure 7 a needs labels.
- o The processes they used to identify new phages is a bit convoluted and hard to understand from the text. They used Kraken classifications, virsorter, comparison to the IMG/VR database and VContact2 based clustering, it seems. Perhaps a main figure could be an outline of the bioinformatic analysis process they used?
- o The discussion section on the CRISPR spacers is interesting, but very lengthy and a bit too speculative given the limited results that are presented. I might recommend scaling this back a bit and highlighting the main take home points.
- o I could not access the supplementary tables.

Minor concerns:

- Line 42-43 – would change the word “main” to “most diverse”
- Line 45-47 – There have been quite a few studies on organisms such as crAssphage; and there is an emerging literature on bacteriophages in the intestinal microbiome. I'd agree that the oral “phageome” has been less studied and would focus the statement on this fact.
- The manuscript is generally well written, but would still benefit from a careful review for English grammar/word usage.
- The authors switch back and forth between reporting number of contigs classified in a certain way (example line 158) and percentages (example line 160) – would maintain a standard approach to this – ideally including the range of numbers and the range of percentages.
- Line 178 – what do the authors mean by “predominant phages” – I'm not sure how they came upon this categorization.
- I would expect that long read sequencing would greatly enable the discovery of prophages – did the authors investigate what long reads added in terms of scaffolding and ability to place a prophage in the proper host genomic context?
- Most use several rounds of short-read polishing after long read assembly – is there a reason that only one run of pilon was used?

- Perhaps I missed it – but are the MetaQuast results provided in the supplement? This is important for the reader.

Reviewers' comments:

Reviewer #1 (Remarks to the Author):

Major comments: The utilization of Kraken as the preferred annotation tool for the short reads is worrisome. Internal and published results have shown that Kraken can increase the number of false positives (1). Kraken also seems to not attempt to discriminate or classify viral sequences (2).

Response: Thank you for your comments. We agree that it is not appropriate to use Kraken for an initial taxonomic profiling of the oral virome. To correct this, we have deleted two sentences in the Methods (after line 502) and two sentences in the Results (after line 104). This change does not affect the structure or logic of our paper because our goal was to identify and analyze novel viral contigs through the assembly of long reads.

Another major concern is the high error rate with PromethION and related sequencing technologies. While the authors attempt to address this issue in lines 345-353, in order to understand how this error rate is affecting virus discovery from their metagenomic datasets, the authors need to include a virus, or a viral mock community in the analysis, and use the same sequencing technology and annotation tool.

Response: The use of a viral mock community to understand how the error rate of long reads affected virus discovery was already conducted in a recent study by Matthew Sullivan's group (<https://peerj.com/articles/6800/>). According to the authors of the study, the nanopore long-read metagenomic approach “was first validated using mock communities where it was found to be as relatively quantitative as short-read methods and provided significant improvements in recovery of viral genomes”. Based on the recent study, we extensively revised the Introduction (line 72-77) to include the following: “Several more limitations of the short-read assembly approaches were specifically addressed in recent studies that aimed to overcome them using long-read sequencing²¹⁻²³. One of these studies²² validated the approach of viral long-read metagenomics via nanopore sequencing using mock communities, and found it to be as relatively quantitative as short-read methods, providing significant improvements in recovery of viral genomes, albeit the high error rates” Considering the validation of nanopore sequencing performed previously, we did not consider it necessary to include the analysis of a mock community in our study.

Lines 100-105: The low abundance of human reads in saliva samples is surprising. Can

the authors very briefly include in this part of the Results section a description of the method and why they think it reduces so significantly the number of human reads?

Response: Based on an additional experiment we performed, the method used and reason why it reduces the number of human reads have been included in the Results section (line 104-124) as “An additional experiment demonstrated that this occurred because, in our protocol, we used the OMNIgene ORAL kit, after which we performed the enzymatic DNA extraction: we collected two additional 1 mL samples of saliva from three of the four healthy volunteers using the OMNIgene ORAL kit or, as an alternative, the *RNAlater* stabilization solution, followed by the same procedures of enzymatic DNA extraction, library preparation, and metagenome sequencing using HiSeq (“2nd experiment” in Table S2). The preprocessing of sequence data revealed that the proportion of human reads was 0.08–0.54% when using the OMNIgene kit compared to 28.12–37.57% when using the *RNAlater* kit (Fig. 1). The OMNIgene ORAL kit had the lytic activity for the existing cells including human cells. Because human cells appeared to be more easily lysed than the bacterial cells under preservation in the OMNIgene ORAL kit, the amount of DNA released from the lysed human cells is likely higher than that released from bacteria. In the enzymatic DNA extraction protocol, the salivary sample is first centrifuged to harvest non-lysed microbial and human cells and viral particles as pellets, which are then subjected to DNA extraction. In the case of salivary samples collected using the OMNIgene ORAL kit, the first centrifugation step may have separated human cell-derived DNA/RNA in the supernatant from the pellet, so the pellet contained almost no human DNA (Fig. 1). In contrast, cells in the salivary samples collected using *RNAlater* were generally not lysed under preservation because of an absence of lytic activity by *RNAlater*. Therefore, the pellet obtained by centrifugation contained mostly intact microbial and human cells and viral particles, which were then subjected to enzymatic lysis to extract whole DNA.”, and presented in a new figure (Figure 1, revised manuscript).

Figure 1

Figure 1A and 1B do not have to be one of the main figures. Since the figure is only showing DNA concentration, and number of reads, results can easily be added to two tables. Also, it is not clear from looking at figure 1A what the sample designations are unless you look at the text and understand that these are duplicates.

Response: To address the point raised by the reviewer, as well the suggestion given by Reviewer 2, we have converted Figure 1 into two supplemental tables (Table S1 and Table S2, shown below). Also, the duplicates of each sample are now indicated as “1st” and “2nd” in the column “Order of two samplings” in Table S1, as shown below.

Table S1. Concentration of the DNA extracted from the saliva samples

Condition	Sample	Storage	Order of two samplings	Concentration (ng/ μ l)	Sequencing
1st experiment	1	OMNIgene	1 st	57.47	HiSeq
			2 nd	78.00	PromethION
	2	OMNIgene	1 st	94.07	PromethION
			2 nd	69.00	HiSeq
	3	OMNIgene	1 st	24.47	HiSeq
			2 nd	61.47	PromethION
	4	OMNIgene	1 st	70.40	PromethION
			2 nd	55.20	HiSeq
2nd experiment	1	OMNIgene	1 st	98.80	HiSeq
		RNA/ater	2 nd	420.00	HiSeq
	2	OMNIgene	1 st	25.60	HiSeq
		RNA/ater	2 nd	357.00	HiSeq
	3	OMNIgene	1 st	3.89	HiSeq
		RNA/ater	2 nd	301.00	HiSeq

measured using Qubit dsDNA HS Assay Kit

Table S2. The amount of sequence data generated

Condition	Sample	Storage	HiSeq	PromethION (all)	PromethION (Q>=7)
1st experiment	1	OMNIgene	55.9	38.8	29.7
	2	OMNIgene	37.2	90.1	74.0
	3	OMNIgene	55.5	65.0	46.7
	4	OMNIgene	37.7	65.8	50.3
2nd experiment	1	OMNIgene	22.0	-	-
		RNA/ater	22.0	-	-
	2	OMNIgene	21.8	-	-
		RNA/ater	20.9	-	-
	3	OMNIgene	25.1	-	-
		RNA/ater	19.8	-	-

in Gb

in Gb

in Gb

"Q>=7" represents a subset of PromethION reads with average quality score >= 7

Information from figure 2A and 2B can also be added into a table for easier interpretation.

Response: We have followed your suggestion as well as that of Reviewer 2 and converted Figure 2 into a supplemental table (Table S3) that presents all the assembly statistics calculated using MetaQuast.

Table S3. Assembly statistics calculated using MetaQuast

Statistics	Sample1 (Flye)	Sample2 (Flye)	Sample3 (Flye)	Sample4 (Flye)	Sample1 (hybridSPAdes)	Sample2 (hybridSPAdes)	Sample3 (hybridSPAdes)	Sample4 (hybridSPAdes)
# contigs (>= 0 bp)	3690	2885	5617	3133	663473	368181	669898	293954
# contigs (>= 1000 bp)	3672	2865	5574	3099	72403	41221	67500	41967
# contigs (>= 5000 bp)	3543	2708	5372	2975	15405	10582	13959	10700
# contigs (>= 10000 bp)	3252	2440	5021	2743	6540	4906	6104	4665
# contigs (>= 25000 bp)	2776	2057	4346	2311	1811	1538	1721	1392
# contigs (>= 50000 bp)	2156	1660	3384	1775	622	554	525	521
Total length (>= 0 bp)	480940349	426707605	615659498	347232509	474528463	316767311	454377538	311183750
Total length (>= 1000 bp)	480927768	426693207	615631595	347206578	359772153	259606689	326354916	252580381
Total length (>= 5000 bp)	480571933	426246616	615030208	346813139	239085332	191724567	214749364	183801403
Total length (>= 10000 bp)	478424393	424302558	612459496	345077886	177731664	152253816	160310919	141798465
Total length (>= 25000 bp)	470276399	418032194	601134271	337900409	107014257	101285312	94179983	92691336
Total length (>= 50000 bp)	447115906	403300884	565205343	318041480	66864758	67721756	53128072	62860338
Largest contig	3631366	5097128	3203214	2660737	992493	1118014	633360	1624546
GC (%)	43.73	43.09	45.11	44.91	46	44.59	46.6	46.16
N50	256159	344722	187441	207879	9733	14661	9663	13023
N75	113520	144594	95912	100380	3495	4749	3373	4543
L50	346	230	786	369	6759	3045	6396	3302
L75	1067	711	1939	980	22804	11194	21300	11883

Lines 187-188: Can the authors expand on what they mean with remote homologs of antimicrobial-resistance genes throughout the text as needed? Is it hits with <95% identity, for example?

Response: As suggested by the reviewer, we checked the amino acid sequence identity and the aligned length of each remote homolog compared to its query sequence in UniProt, by looking at the output of HHblits. To further clarify what we mean by remote homologs we added the following sentence to the Results section of “*Streptococcus* phage/prophage group and genes for antimicrobial resistance and integrase” (line 234-238): “The remote homologs showed an average percentage of amino acid sequence identity of 46% (maximum 99%, minimum 20%, interquartile range 28-61%) and an average percentage of aligned length of 66% (maximum 95%, minimum 13%, interquartile range 47-88%), compared to corresponding amino acid sequences in UniProt database”. We have added a similar sentence to the Results section that refers to “Jumbo phages/prophages” (line 278-282).

Figure 5 and lines 190-193: It is suggested that this is a prophage, but the highlighted annotation does not show markers of lysogeny, particularly integrases or any genes part of lysogeny modules. Can the authors expand on the detection of integrases, etc. in the presumptive prophages?

Response: Yes, we detected integrase genes using HHblits, and colored them in orange in the revised figure, as shown below (the figure is now part of the supplemental data [Figure S7] as requested by Reviewer 2). The figure shows a typical example in which integrase genes are located at the end of the prophage. As suggested, we expanded on the detection of integrases and added the following sentences to the Results section of “*Streptococcus* phage/prophage group and genes for antimicrobial resistance and integrase” (line 243-252): “The high coverage *Siphoviridae* prophage in Fig. S7 is a

typical example in which integrase genes are located at the end of the prophage (colored in orange). Similar to the antimicrobial resistance genes, we searched for remote homologs of integrase genes in the group of 28 *Siphoviridae* phages/prophages, and they were detected in 46.4% of them. Average percentage of amino acid sequence identity was 40% (maximum 70%, minimum 21%, interquartile range 25-59%) and average percentage of aligned length was 91% (maximum 100%, minimum 70%, interquartile range 90-95%), compared to corresponding amino acid sequences in UniProt database. If this analysis was extended to all of the high coverage (top 10%) viral sequences predicted by VirSorter, the remote homologs of integrase genes were detected in 52.5% (85 out of 162, indicated in “Integrase” column in Table S4) across the 4 samples.”

Methods: Can the authors specify in the methods if they used a DNase treatment before nucleic acid extraction? If not, it is highly possible that host DNA may have been analyzed along with the viral sequences. That may explain plasmid-like sequences in the presumptive prophage sequenced.

Response: We did not perform DNase treatment, as explained in a new figure (Figure 1, revised manuscript). We agree with the possibility pointed out by the reviewer, and in order to assess it, we used PlasFlow to predict the presence of plasmid sequences in all viral sequences predicted by VirSorter. The analysis revealed that 1.9%-4.5% of the viral sequences were predicted to belong to plasmids.

We have also added the following sentence in the Results (line 268-272): “, although it was not predicted to be a plasmid by a machine-learning program PlasFlow ⁴⁰, perhaps

because of absence of a similar plasmid sequence in the reference training database. Proportion of predicted plasmids among all the predicted viral sequences was 1.9-4.5% (“Plasmid? (PlasFlow)” column in Table S4).

Minor:

Line 264: Replace “the on an” with “that an”

Response: We thank the reviewer for calling our attention to this error. We have corrected it accordingly.

Reference titles:

1. Phylogenetic microbiota profiling in fecal samples depends on combination of sequencing depth and choice of NGS analysis method.
2. VirFinder: a novel k-mer based tool for identifying viral sequences from assembled metagenomic data.

Reviewer #2 (Remarks to the Author):

In this manuscript, Yahara and colleagues perform metagenomic sequencing on oral samples (human saliva) using long read sequencing. Applying this fairly new technology, the authors generate >30Gb data (a combination of nanopore and HiSeq data) and then perform assembly. They queried the assembly data to identify viral contigs and by comparing these data to available published data find that a substantial proportion of the contigs are candidate novel phages. Some of these phages encode potential remote homologs of antimicrobial resistance genes.

Overall, the manuscript presents some nice, new data (oral long-read sequencing) from a very limited number of human samples (four subjects, in total). The focus on the oral phageome is interesting and the bioinformatic analyses are generally appropriate. There isn't quite enough methodological detail to understand how the bioinformatic pieces fit together, and in some cases, more methodological detail is needed to evaluate the choices made (e.g. what kraken database was used and how cutoffs were selected). The identification of novel jumbophages is very interesting and was probably the highlight of the manuscript. While interesting, the manuscript could benefit from improvements in writing flow, figure presentation (both how the figures are presented and what is presented in the main manuscript), and much better referencing of the current status of the field, as well as more accurate representations of what is known in the field. I have outlined several major and minor concerns below, which if addressed, would increase my enthusiasm for this piece of work.

Response: We thank the reviewer for the insightful and constructive comments. Please find below our answers to each of your concerns.

Major concerns:

- In general, the introduction, while interesting, makes some critical misinterpretations or misstatements about the current status of the field or prior literature. There are also some notable omissions. I've highlighted some of them here:

o Line 69 – in the PHASTER paper that the authors refer to, the manuscript cited showed that sensitivity of PHASTER increases to >90% in CONTIGS of >20kb – NOT contigs assembled from reads > 20kb. Thus, a core logical point that the authors point at to justify their application of long reads is flawed.

Response: We thank the reviewer for pointing out this important mistake. To fix it, we have removed the mentioned sentence (after line 68) and the text has been rewritten to

express the study's core rationale according to your additional comments mentioned below.

o Lines 75 onward – many have shown that PacBio sequencing can be performed on as little as 100ng of DNA (example: doi: 10.3390/genes10010062). Certainly gut microbiome-based libraries can be made with 1ug of DNA (see <https://www.nature.com/articles/s41587-020-0422-6>)

o Several papers have come out showing the utility of nanopore sequencing for microbiome studies (none of which are cited in this manuscript) – notably:

♣ <https://www.nature.com/articles/s41587-020-0422-6>

♣ <https://www.nature.com/articles/s41587-019-0202-3>

♣ <https://www.ncbi.nlm.nih.gov/pubmed/31359005>

o I don't think that the statement that nanopore library prep requires less DNA than PacBio is accurate based on the available literature. That said, it is a useful platform and there is plenty of justification to use nanopore.

Response: We agree with all the reviewer's comments, and have incorporated them in the revised manuscript by rewriting the mentioned paragraph (line 68-81) as follows:

“These previous studies, however, were based on short-read sequencing data generated using Illumina sequencer. However, the short-read assembly approaches do have limitations, particularly in assembly contiguity^{16,17}. Specifically, generation of short fragmented assemblies impedes the analysis of genomic context or detection of viral sequences using programs, such as VirSorter^{18,19}, which requires long genomic fragments with sufficient evidence to warrant a prediction²⁰. Several more limitations of the short-read assembly approaches were specifically addressed in recent studies that aimed to overcome them using long-read sequencing²¹⁻²³. One of these studies²² validated the approach of viral long-read metagenomics via nanopore sequencing using mock communities, and found it to be as relatively quantitative as short-read methods, providing significant improvements in recovery of viral genomes, albeit the high error rates. More recent shotgun metagenomic analyses using nanopore long-reads demonstrated improved assembly contiguity^{16,17}, with much less fragmented assemblies than were achieved by PacBio sequencing, possibly due to less variable coverage with nanopore sequencing¹⁷, although the sequencing error rates are lower in PacBio compared to nanopore²⁴”

o In lines 98-99 – how were samples multiplexed for the PromethION sequencing? Did

they obtain 38.8Gb to 90.1Gb Per sample? It is not clear to me based on the way the manuscript was written, although based on Figure 1b, it appears that this is the case.

Response: Yes, we obtained 38.8Gb to 90.1Gb per sample. To make this clearer in the text, we have added “per sample” to the sentence.

o Figure 1 – this can likely be supplemental.

Response: Figure 1 is now provided as supplemental data and in the form of a table (Table S1) as suggested by Reviewer 1.

o Line 102-104 – I agree that the % of human reads is very surprisingly small – how does this compare to other studies that used this extraction protocol? Human cells are typically much easier to lyse than bacterial cells...

Response: There is no study that has used the same enzymatic DNA extraction protocol for human saliva samples and reported % of human reads. We agree with the reviewer that human cells are typically much easier to lyse than bacterial cells. As explained above to Reviewer 1, we performed an additional experiment that revealed that the surprisingly small percentage of human reads occurred because, in our protocol, we used the OMNIgene ORAL kit that has the lytic activity, after which we performed the enzymatic DNA extraction (Figure 1, revised manuscript).

o Line 108 – need more detail re what the “full database” for Kraken is. The manuscript that is referred to was published in 2014 and there is a minidatabase that is described with it in addition to larger builds. The databases are constantly being updated and many users create their own custom database. Please provide more detail on the exact database used in the methods.

Response: We created the “full database” according to a procedure available at https://github.com/mw55309/Kraken_db_install_scripts. As explained in this website, the procedure is very similar to “adding a custom database” as outlined in Kraken’s manual. However, we agree with Reviewer 1 and consider Kraken inappropriate for an initial taxonomic profiling focusing on the oral virome. Thus, in the revised manuscript, we have deleted the two sentences in Methods section (after line 112) and the two sentences in the Results section that mentioned Kraken (after line 477).

o The method applied for taxonomic classification described in lines 139 onward is interesting – but how did they validate this approach? Is there precedent for this? How was the cutoff value decided?

Response: The Contig Annotation Tool (CAT) for the taxonomic classification of contigs was validated by its developers (von Meijenfeldt et al., 2019; cited in our manuscript) using simulated contig datasets, in which the cutoff value was decided in order to achieve a balance between the classification precision and the fraction of classified sequences. Decreasing the cutoff value in classifications based on fewer ORFs leads to more tentative classifications at lower taxonomic ranks, which results in more sequences being classified at lower taxonomic ranks, albeit rendering a lower precision. To clarify how the cutoff value was decided, we added the following information to the paragraph (line 162-163): “which was decided to achieve a balance between the classification precision and fraction of classified sequences.”

o Figure 2 – can be supplemental and probably best represented as a table.

Response: In the revised manuscript, Figure 2 is now represented as a new supplemental table (Table S1), that includes all statistics calculated using MetaQuast.

o Figure 3 – What do the numbers of the virsorter categories represent (1, 2, 4, 5)? I don't understand that. Also, this figure might be better if separated into panels for the pro-phage and phages separately. Or perhaps this might be better as a table instead of a figure.

Response: The numbers represent the following VirSorter categories: “most confident phage” (1); “likely phage” (2); “most confident prophage” (3); and “likely prophage” (4). As suggested, we converted Figure 3 into a new table (Table 1, below) with separate columns for phages and prophages, and in which category numbers have been replaced by a “most confident” or a “likely” tag.

Table 1. The number and proportion of novel viral sequences identified in each sample and stratified by the “most confident” and “likely” phages and prophages

Sample	Phage				Prophage			
	most confident		likely		most confident		likely	
	novel	known	novel	known	novel	known	novel	known
1	0 (0%)	5 (100%)	54 (59%)	37 (41%)	26 (46%)	30 (54%)	233 (74%)	83 (26%)
2	0 (0%)	7 (100%)	37 (49%)	38 (51%)	27 (56%)	21 (44%)	205 (72%)	81 (28%)
3	7 (44%)	9 (56%)	63 (52%)	58 (48%)	19 (37%)	33 (63%)	323 (77%)	97 (23%)
4	1 (20%)	4 (80%)	25 (42%)	35 (58%)	3 (12%)	21 (88%)	73 (56%)	58 (44%)

The novel phages and prophages do not cluster with any viral sequence in the IMG/VR v2.0 database

o The discovery of the oral microbiome jumbo phages is interesting – table 1 is excellent.

Response: We are grateful for your appreciation.

o I don't find the bottom panel of Figure 4 terribly informative.

Response: We deleted the bottom panel of Figure 4 (currently Figure 3) and improved the legend boxes to remove unnecessary information and make its order (outer ring followed by inner ring) correspond to the figure.

o Figure 5 & 6 should be supplementary.

Response: We have moved both figures to supplemental material.

o Figure 7 a needs labels.

Response: Thanks for pointing out this mistake. We have fixed it in the revised version (Figure 4, shown below). The x-axis is now labeled as the “Number of genes” and the y-axis is now labeled as the “Num. of individuals (out of the four) sharing the gene.”.

o The processes they used to identify new phages is a bit convoluted and hard to understand from the text. They used Kraken classifications, virstorter, comparison to the IMG/VR database and VContact2 based clustering, it seems. Perhaps a main figure could be an outline of the bioinformatic analysis process they used?

Response: We understand the reviewer's concern. To address it, we have created a new supplemental figure (Figure S1, shown below) that outlines the bioinformatics analysis

done.

o The discussion section on the CRISPR spacers is interesting, but very lengthy and a bit too speculative given the limited results that are presented. I might recommend scaling this back a bit and highlighting the main take home points.

Response: Following the reviewer’s suggestions, we have removed three sentences from the mentioned section. The main take-home message now reads: “it suggests that the oral phages currently present in human saliva could be under selective pressure escaping from CRISPR immunity” (line 428-429); and is followed by: “as suggested in a recent review ⁴ and study ¹⁵” Moreover, the paragraph now highlights another take-home message, expressed in a concluding sentence, as follows: “Therefore, the results presented in our study are probably more general and reliable than those obtained in the previous study, with regard to the understanding of the overall relationship between CRISPR spacers and protospacers in the oral microbiota.”

o I could not access the supplementary tables.

Response: We have uploaded the supplemental tables through the manuscript submission system of this journal. If you cannot access them, please use the following direct link:

https://www.dropbox.com/s/kgijw6s66zoq3aq/PromethION_neovirology_manuscript_SI_files.zip?dl=0

Minor concerns:

- Line 42-43 – would change the word “main” to “most diverse”

Response: We have made the suggested change in the revised manuscript.

- Line 45-47 – There have been quite a few studies on organisms such as crAssphage; and there is an emerging literature on bacteriophages in the intestinal microbiome. I'd agree that the oral “phageome” has been less studied and would focus the statement on this fact.

Response: We thank the reviewer for this suggestion. We have modified the sentence as follows: “bacteriophages (phages), or bacterial viruses, in the intestinal microbiome have received increasing attention over the last decade (5, 6), whereas those present in the oral microbiota have been less studied.”

- The manuscript is generally well written, but would still benefit from a careful review for English grammar/word usage.

Response: We have conducted a careful review of the manuscript, focusing on English grammar/word usage, with the help of a professional English editing service by Editage.

- The authors switch back and forth between reporting number of contigs classified in a certain way (example line 158) and percentages (example line 160) – would maintain a standard approach to this – ideally including the range of numbers and the range of percentages.

Response: Thank you for calling our attention to this lack of consistency. To rectify it, we have modified sentences containing these inconsistencies to include both the range of numbers and the range of percentages of contigs or viral sequences. During this process, we found some errors in the expression of the results relative to vConTACT v2.0. All these errors are now fixed in the revised manuscript.

- Line 178 – what do the authors mean by “predominant phages” – I'm not sure how they came upon this categorization.

Response: We have replaced “predominant” by “high coverage” in that sentence.

- I would expect that long read sequencing would greatly enable the discovery of

prophages – did the authors investigate what long reads added in terms of scaffolding and ability to place a prophage in the proper host genomic context?

Response: We thank the reviewer for this insightful observation which prompted us to perform additional analyses to investigate the information added by long reads on the scaffolding and genomic context of prophages. In the revised manuscript we have included these new results in a section titled **“Enhanced scaffolding and placing a prophage in its host genomic context”** and in Figure 2, as shown below:

Enhanced scaffolding and placing a prophage in its host genomic context. Among the known “most confident” and “likely” prophages clustered with a viral sequence in the IMG/VR v2.0 database, we found cases in which scaffolding and its host genomic context were much improved compared to the viral sequences assembled from short-reads. An example is shown in Fig. 2A, in which a high coverage prophage with 77.4kb embedded in a 674.7kb contig with on average 444x short-read coverage has a genomic region in the middle aligned with 11.1kb viral sequence in the IMG/VR v2.0 database without information of predicted host. At the end of the high coverage prophage assembled from the long-reads, there is a CDS encoding integrase (depicted as an orange arrow). An enlarged genome map of the prophages is shown in Fig. S6, in which genes were characterized by the HMM-based iterative protein searches. Genes for phage morphogenesis (colored in brown) are clustered in the region (approximately from 48kb to 60kb, indicated by a purple horizontal line) corresponding to the aligned region in Fig. 2A. At the right of the end of prophage, there is a CDS encoding enolase (depicted as a green arrow), which is a surface-exposed adhesion protein of *Streptococcus* suggested to be a phage receptor or to interact with proteins of phages³³. The aligned viral sequence in the IMG/VR v2.0 database encoded 14 CDSs, but its taxonomic classification using CAT was impossible. In contrast, the prophage and host sequences at the both ends were all assigned to *Streptococcus* genus with the score 0.74, 0.90, and 0.79, respectively.

Distribution of differences in length between each of the known “most confidence” and “likely” prophages and corresponding viral sequences in the IMG/VR v2.0 database clustered with the prophages by genome clustering³⁰, which is shown in Fig. 2B. Positive values indicate that the prophages assembled from long-reads are longer than the corresponding viral sequences assembled from the short-reads. The average, median, and interquartile range (IQR) was 47.4 kb, 39.2 kb, and 24.8-60.5 kb, respectively. Only 2.8% (11 out of 393) of the prophages assembled from the long-reads were shorter (i.e., x-axis of Fig. 2B < 0) than the corresponding viral

sequence assembled from the short-reads.

B. Differences in length among the all known "most confidence" and "likely" prophages

- Most use several rounds of short-read polishing after long read assembly – is there a reason that only one run of pilon was used?

Response: The reason we used a single run of pilon is that running pilon on all contigs (1.87Gb in total) is computationally intense and can take several weeks. Although the computation time depends on the number of CPU cores available at the time, the super computer that we routinely use was less available than it was a year ago, thereby making multiple rounds of polishing with pilon impossible.”

- Perhaps I missed it – but are the MetaQuast results provided in the supplement? This is important for the reader.

Response: We have provided the MetaQuast results in a supplemental table (Table S1), which presents MetaQuast statistics.

REVIEWERS' COMMENTS

Reviewer #1 (Remarks to the Author):

The authors have answered the concerns brought in the previous round of revisions and have addressed other suggestions as needed. I believe the manuscript is now suitable for publication.

Tasha M. Santiago-Rodriguez

Reviewer #2 (Remarks to the Author):

The authors have done a very good job of revising the manuscript and have largely addressed my comments. The revised manuscript is much improved.

My only remaining comment is that many of the figures I received are fairly low resolution - I recommend attempting to improve them for easier viewing and more consistent visual appeal (consistent fonts, sizes/etc).

Reviewers' comments:

Reviewer #1 (Remarks to the Author):

The authors have answered the concerns brought in the previous round of revisions and have addressed other suggestions as needed. I believe the manuscript is now suitable for publication.

Tasha M. Santiago-Rodriguez

Reviewer #2 (Remarks to the Author):

The authors have done a very good job of revising the manuscript and have largely addressed my comments. The revised manuscript is much improved.

My only remaining comment is that many of the figures I received are fairly low resolution - I recommend attempting to improve them for easier viewing and more consistent visual appeal (consistent fonts, sizes/etc).

Response: We have improved the figures by increasing the resolution, making them vectorial, and using consistent fonts and sizes.